# Aquaporin-3 regulates endosome-to-cytosol transfer via lipid peroxidation for cross presentation

**Sam C. Nalle**[1], **Rosa Barreira da Silva**[1], **Hua Zhang**[2], **Markus Decker**[1], **Cecile Chalouni**[1], **Min Xu**[2], **George Posthuma**[3], **Ann de Mazière**[3], **Judith Klumperman**[3], **Adriana Baz Morelli**[4], **Sebastian J. Fleire**[5], **Alan S. Verkman**[6], **E. Sergio Trombetta**[7], **Matthew L. Albert**[1], **Ira Mellman**[1]*

1 Department of Cancer Immunology, Genentech, South San Francisco, California, United States of America, 2 Department of Immunology, Genentech Inc., South San Francisco, California, United States of America, 3 Department of Cell Biology, University Medical Center Utrecht, Utrecht, The Netherlands, 4 CSL Ltd., Parkville, Victoria, Australia, 5 New York University Cancer Institute, New York, New York, United States of America, 6 Department of Medicine, University of California, San Francisco, California, United States of America, 7 Boehringer Ingelheim, Ridgefield, Connecticut, United States of America

* mellman.ira@gene.com

**Data Availability Statement:** All relevant data are within the paper and its supporting information files.

**Funding:** The authors SCN, RBS, MD, CC, MLA and IM are employed by Genentech. The author

## Abstract

Antigen cross presentation, whereby exogenous antigens are presented by MHC class I molecules to CD8+ T cells, is essential for generating adaptive immunity to pathogens and tumor cells. Following endocytosis, it is widely understood that protein antigens must be transferred from endosomes to the cytosol where they are subject to ubiquitination and proteasome degradation prior to being translocated into the endoplasmic reticulum (ER), or possibly endosomes, via the TAP1/TAP2 complex. Revealing how antigens egress from endocytic organelles (endosome-to-cytosol transfer, ECT), however, has proved vexing. Here, we used two independent screens to identify the hydrogen peroxide-transporting channel aquaporin-3 (AQP3) as a regulator of ECT. AQP3 overexpression increased ECT, whereas AQP3 knockout or knockdown decreased ECT. Mechanistically, AQP3 appears to be important for hydrogen peroxide entry into the endosomal lumen where it affects lipid peroxidation and subsequent antigen release. AQP3-mediated regulation of ECT was functionally significant, as AQP3 modulation had a direct impact on the efficiency of antigen cross presentation *in vitro*. Finally, *AQP3*[-/-] mice exhibited a reduced ability to mount an anti-viral response and cross present exogenous extended peptide. Together, these results indicate that the AQP3-mediated transport of hydrogen peroxide can regulate endosomal lipid peroxidation and suggest that compromised membrane integrity and coordinated release of endosomal cargo is a likely mechanism for ECT.

ABM is employed by CSL Ltd.. The author EST is employed by Boehringer Ingelheim. These funders provided support in the form of salaries for authors SCN, RBS, HZ, MD, CC, MX, ABM, EST, MLA, and IM, but did not have any additional role in the study design, data collection and analysis, decision to publish, or preparation of the manuscript. The specific roles of these authors are articulated in the 'author contributions' section.

**Competing interests:** The authors SCN, RBS, MD, CC, MLA and IM are employed by Genentech. The author ABM is employed by CSL Ltd.. The author EST is employed by Boehringer Ingelheim. The commercial affiliation of the authors does not alter our adherence to PLOS ONE policies on sharing data and materials. There are no patents, products in development or marketed products to declare.

## Introduction

The cross presentation of antigen is required in order to generate an effective cytotoxic CD8 + T cell (CTL) response to viruses, bacteria, parasites, and cancer [1, 2]. Whereas CTLs can be primed by direct presentation on MHC class I, cross presentation allows for the generation of adaptive immunity to exogenous antigens not expressed by antigen presenting cells (APCs). Although some cross presentation can occur within the phagosome itself [3–5], it is far more common for cross presented antigen to escape from phagosomes to be processed in the cytosol by proteasomal proteolysis with the resulting peptides transported into the ER via the TAP1/ TAP2 complex and loaded onto MHC class I molecules [6, 7]. Owing partly to phagosomes characterized by higher pH, reduced enzymatic activity, and antigen preservation [8–11], DCs are particularly efficient at endosome to cytosol transfer (ECT) of antigen [12, 13], yet how this process occurs remains a mystery. One view is that antigens cross the endosome membrane by a process analogous to the retrotranslocation of content from the ER lumen for degradation in the cytosol (ERAD) [14, 15]. However, exhaustive proteomic analysis of isolated endocytic organelles has failed to identify likely candidates [16, 17] and definitive genetic evidence is lacking [18]. In addition, recent work demonstrated that the reduced cross presentation observed when the ERAD transporter Sec61 was inhibited was not due to a decrease in ECT, as was originally suggested, but rather correlated with a decrease in the expression of the machinery necessary for MHC class I presentation in general [19].

Another mechanistic explanation for ECT could be a coordinated, but more generalized process of antigen escape. Indeed, evidence in support of this mechanism was provided by a recent study focused on the role of phagosomal NOX2 and the generation of ROS in the phagosomal lumen, resulting in lipid peroxidation and antigen release [20]. Although this is a key observation, there are several important questions that remain: namely, is NOX2-dependent endosomal ROS the only source of free radicals necessary to induce lipid peroxidation and antigen release? And, as *Cybb*$^{-/-}$ (NOX2-deficient) mice do not have cross presentation defects *in vivo*, what other mechanisms are important for ROS generation and endosomal lipid peroxidation? Here, we provide evidence that ECT is coupled to the pathogen-triggered release of mitochondrial ROS (mROS), which may then mediate lipid peroxidation and subsequent membrane disruption following enhanced uptake of $H_2O_2$ by the endosomal peroxide channel aquaporin-3.

## Materials and methods

### Materials

The pIRES-DsRed2 and pDsRed2-C1 vectors used to express various human aquaporin constructs were from Clontech. The human pCMV6-AC-AQP9-GFP expression vector was from Origene. 30% $H_2O_2$ solution, TTFA, epoxomicin, and OVA (Grade VI, filtered) were purchased from Sigma-Aldrich. Peptidoglycan (PGN) and pure lipopolysaccharide (LPS) were purchased from Invivogen. Recombinant HIV1 p24 was purchased from Abcam. All flow cytometry analysis was performed on a FACS Canto II (BD Biosciences) and data was further analyzed using FlowJo (Tree Star). Fixed and live cell confocal imaging were respectively performed using a Leica SP5 microscope and a Leica SP8 microscope dotted with a Ludin environmental chamber and $CO_2$ controller. A recombinant β-lactamase-DHFR fusion protein, consisting of a FLAG-tagged N-terminal *E. coli* β-lactamase and C-terminal human DHFR chimera, was expressed in *E. coli* using the pET vector backbone (Novagen). rHyPer was created by *E. coli* expression of a FLAG-tagged version of *E. coli* OxyR protein using the pET vector backbone.

## Animals and cell culture

Research involving animals complied with protocols approved by the Genentech Institutional Animal Care and Use Committee. All efforts were made to alleviate suffering and minimize the number of animals needed for each study. Animals were euthanized under deep isofluorane-induced anesthesia, followed by cervical dislocation. C57BL/6 mice (Charles River) were used for all experiments, except when analyzing $AQP3^{-/-}$ mice on the CD-1 background [21], in which case littermate $AQP3^{+/+}$ WT controls were used. B6.129S6-$Cybb^{-/-}$ (NOX2-deficient) mice were purchased from Jackson Laboratories. BMDCs were differentiated from bone marrow of 6–12 week old mice using 10 ng/ml GM-CSF (Peprotech) and 5 ng/ml IL-4 (Life Technologies), and evaluated on day 6 of culture. For experiments involving flow cytometry analysis of BMDCs (e.g. ECT assay, phagocytosis measurements, ROS analysis), cells were labeled with CD11c (BD Pharmingen) antibody and the CD11c+ population was analyzed. For isolation of splenic DCs, T, B, and NK cells were depleted using the Miltenyi CD8+ Dendritic Cell Biotin-Antibody Cocktail, followed by positive selection using CD11c microbeads (Miltenyi). The cells were rested overnight in 2 ng/ml murine GM-CSF (Peprotech), used in CCF4 and uptake assays, then labeled with CD11c (BD Pharmingen), CD8a (eBioscience), or XCR1 (Biolegend) antibodies and analyzed by flow cytometry.

## RNA-Seq

RNA from flow-cytometry sorted BMDCs was isolated using the RNeasy kit (Qiagen), per the manufacturer's instructions. PolyA RNA-Seq was performed by Otogenetics (Norcross, GA) using an Illumina HiSeq sequencer. Cufflinks was used to carry out differential gene expression analysis.

## ECT assays

For the CCF4 ECT system, cells were loaded with 2 μM CCF4-AM (Life Technologies) substrate for one hour at room temperature. After 3 washes, recombinant β-lactamase (VWR) was added, and the cells were incubated at 37°C for 30 minutes when examining splenic DCs or 1 hour when examining BMDCs or HEK293 cells. For experiments using TTFA in BMDCs, the β-lactamase incubation was 30 minutes. For experiments with ISCOMATRIX™ adjuvant (CSL Ltd.), the β-lactamase incubation took place in a media solution containing 10% ISCOMATRIX™ adjuvant. Data is presented as cleaved substrate median fluorescence intensity (MedFI), except with HEK293 cells. Overexpression of WT AQP3 also increased phagocytosis of fluorescently-labeled β-lactamase, therefore, HEK293 CCF4 data is presented as "ECT efficiency," whereby the MedFI of the cleaved substrate signal is divided by the MedFI of internalized β-lactamase. For the GAL4 ECT system, siRNA from Dharmacon and Qiagen was used. For the initial siRNA screen, HEK293s stably expressing the GAL4-UAS reporter were transfected with different siRNA in individual wells of a 96 well plate. 24 hours later, 5 μg/ml recombinant GAL4-TA (Sigma-Aldrich) was added for an additional 14 hours, followed by analysis by fluorescence microscopy. The analysis of GAL4 ECT in BMDCs was performed by retroviral transduction of the GAL4-UAS reporter in WT BMDCs, followed by transfection of siRNA on day 5 of culture. 24 hours later, 10 μg/ml GAL4-TA was added for 5 hours, washed away, and then cells were incubated for an additional 14 hours. Reporter fluorescence was analyzed by flow cytometry.

## Immunofluorescence and phagosome association analysis

On day 6 of culture, WT and $AQP3^{-/-}$ BMDCs were fixed in 4% PFA and permeabilized with 0.05% saponin, followed by detection with anti-AQP3 (Alomone Labs) and Hoechst 33342 (Life Technologies). HEK293 cells stably expressing fluorescently-tagged AQP3 or AQP3

  

2xLLmut were incubated with 100:1 (beads:cells) ratio of fluorescently-labeled OVA-coated 1 μm tosylactivated Dynabeads (Life Technologies) for 1 hour. Cells were homogenized (Isobiotec) and analyzed by flow cytometry.

## Electron microscopy

HEK293 cells stably expressing fluorescently-tagged AQP3 (an AQP3-DsRed2 fusion protein) were incubated with 1 μm latex beads (Polysciences) for 2 hours prior to fixation with 2% para-formaldehyde (PFA)/0.2% glutaraldehyde. Fixed cells were processed for immuno-EM as described earlier [22]. Briefly, fixed cells were embedded in 12% gelatin, cryoprotected with 2.3 M sucrose, and frozen in liquid nitrogen. Cryosections, cut at -120˚C and picked up with 1% methylcellulose in 1.2 M sucrose, were labeled with anti-RFP (Rockland) followed by gold-conjugated Protein A (Cell Microscopy Core, Utrecht, the Netherlands). The sections were contrasted with a 1.8% methylcellulose, 0.4% uranyl acetate mixture and examined with a Tec-nai T12 (FEI) electron microscope.

## ROS, mitochondrial superoxide, phagosomal $H_2O_2$, LysoSensor, and lipid peroxidation measurements

For ROS measurements, cells were loaded with 2 μM CM-$H_2$DCFDA (Life Technologies) for 30 minutes at room temperature. After 2 washes, stimuli were added for 15 minutes in media at 37˚C followed by analysis by flow cytometry. Mitochondrial superoxide was measured using MitoSOX Red (Life Technologies). BMDCs were incubated with 5 μM MitoSOX Red for 30 minutes at room temperature. After 2 washes, β-lactamase was added for 15 minutes in media at 37˚C followed by analysis by flow cytometry. Phagosomal $H_2O_2$ content was assessed using 2 mg/ml recombinant HyPer (rHyPer) following a 5 minute incubation. In the presence of $H_2O_2$, the conformational change in the HyPer protein results in an increase in FITC signal with a corresponding decrease in AmCyan signal when analyzed by flow cytometry. To calculate the $H_2O_2$ ratio, the ratio of the MedFI of FITC divided by the MedFI of AmCyan in the presence of rHyPer incubation was subtracted by the background FITC/AmCyan ratio of untreated (no rHyPer incubation) cells. For LysoSensor analysis, transfected and sorted HEK293s were incubated with 1 μM LysoSensor Green DND-189 (Life Technologies) for 30 minutes at 37˚C, followed by analysis by flow cytometry. Phagosomal lipid peroxidation was analyzed by co-incubating BMDCs with a 50:1 ratio of fluorescently-labeled OVA-coated 1 μm Dynabeads and 10 μM C11-bodipy lipid peroxidation indicator (Life Technologies) for 1.5 hours. Cells were then homogenized and analyzed by flow cytometry. The phagosomal lipid peroxidation ratio was determined by gating on OVA+ beads and calculating the C11-bodipy ratiometric fluorescent shift indicative of oxidation.

## Measurement of DHFR inhibition by MTX

DHFR activity of the recombinant β-lactamase-DHFR fusion protein was evaluated using the DHFR Assay Kit (Sigma-Aldrich), following the manufacturer's instructions. In the cell-free DHFR assay and the CCF4 ECT experiments, β-lactamase-DHFR was exposed to 500 nM MTX (Sigma-Aldrich) for 5 minutes at room temperature prior to initiating analysis.

## Phagocytosis measurements

Alexa Fluor (AF)-488-labeled or AF-647-labeled β-lactamase or OVA were incubated for 30 minutes at the indicated doses. 1 μg/ml AF-647-labeled αDEC205 antibody was loaded at 4˚C for 20 minutes, washed, and then incubated for an additional 30 minutes at 37˚C.

## BMDC viral transduction

293T cells were used to generate retrovirus. Lin- hematopoietic progenitors were isolated from bone marrow from WT mice using a lineage depletion kit (Miltenyi). Lin- progenitors were transduced with retroviral supernatants by spinfection followed by incubation for 3.5 hours at 32˚C. Transduction media was removed, replaced with media containing 50 ng/ml SCF (Peprotech), 10 ng/ml GM-CSF, and 5 ng/ml IL-4, and cells were incubated for 2 days at 37˚C. On day 2, transduced progenitors were sorted by flow cytometry and replated in GM-CSF/IL-4. Cultures received fresh GM-CSF/IL-4 media on day 5 and were analyzed on day 6. *AQP3* knockdown efficiency was 93%, as determined by real-time PCR.

## Antigen presentation assays

For soluble OVA antigen presentation studies, $5x10^4$ transduced BMDCs were incubated with OVA or 1 μM SIINFEKL for 4 hours, fixed with 1% PFA, and then incubated with $2x10^5$ purified CFSE-labeled (Life Technologies) OT-I CD8+ T cells (Miltenyi CD8+ T cell isolation kit) or OT-II CD4+ T cells (Miltenyi CD4+ T cell isolation kit) for 64 hours. T cell division was assessed by CFSE dilution using flow cytometry. For cross presentation experiments with ISCOMATRIX™ adjuvant, $2x10^5$ BMDCs were incubated with OVA in media that did or did not contain 10% ISCOMATRIX™ adjuvant for 4 hours, fixed with 1% PFA, and then incubated with $2x10^5$ purified CFSE-labeled OT-I CD8+ T cells. For antibody-delivered cross presentation studies, $5x10^4$ transduced BMDCs were incubated with 1 μg/ml αDEC205-OVA for 1 hour, washed twice, and incubated with $2x10^5$ purified CFSE-labeled OT-I CD8+ T cells for 48 hours. T cell division was assessed by CFSE dilution using flow cytometry.

## In vivo experiments

For LCMV, LCMV Armstrong stocks were prepared and quantified as previously described [23]. Mice (CD-1 background) were infected intravenously with $2x10^6$ plaque-forming units (PFU). 8 days after infection, mice were euthanized and tissue was analyzed. For LCMV antigen-specific IFNγ producing cells, splenocytes were isolated, stimulated with 1 μg/ml LCMV NP$_{118-126}$ peptide for 1 hour and then in the presence of GolgiPlug (BD Biosciences) and peptide for an additional 4 hours, followed by staining with rat anti-mouse IFNγ (eBiosciences) using eBiosciences intracellular staining reagents. For viral titer, monolayers of MC57 cells were infected with serially diluted tissue homogenates. 72 hours after infection, the cells were fixed with 4% paraformaldehyde and permeabilized with 0.5% Triton-X. Viral plaques were stained with anti-LCMV nucleoprotein (anti-LCMV NP, clone VL4) and HRP conjugated anti-rat IgG (Millipore) and visualized with O-phenylenediamine (OPD, Sigma). For anti-OVA Ig experiments, $AQP3^{-/-}$ or WT control mice were injected i.p. with 100 μg of OVA in 100 μl at a 1:1 volume ratio with Complete Freund's Adjuvant (CFA) from Pierce. 10 days later, mice were sacrificed and blood was collected. Serum was analyzed for anti-OVA Igs (A+G+M) by ELISA (Alpha Diagnostic), following the manufacturer's instructions.

For extended peptide immunizations, $AQP3^{-/-}$ mice (CD-1 background) and WT controls were injected intraperitoneally (i.p.) with 50 μg of a synthesized 28 amino acid extended peptide (408–435) from the rat HER-2/neu protein, 25 μg poly I:C (Invivogen) and 25 μg anti-CD40 (clone FGK45). 1 week later, mice were sacrificed, splenocytes were harvested, and antigen-specific CD8+ T cells were determined using the H-2D(q)/rat HER-2/neu$_{420-429}$ tetramer obtained from the National Institute of Allergy and Infectious Diseases MHC Tetramer Core Facility (Atlanta, GA).

## Results

### Two independent functional genomics screens identify aquaporin-3 (AQP3) as a regulator of ECT

To identify genes that might be involved in transferring internalized protein antigens from endosomes to the cytosol, we adapted a well-characterized β-lactamase-CCF4 assay to assess ECT as a function of gene expression level [12, 13, 15]. Bone marrow-derived dendritic cells (BMDCs) were loaded with a cytosolic fluorescent substrate that is cleaved by β-lactamase following its escape from endosomes (Fig 1A). Live BMDCs were then sorted based on high levels of cleaved substrate ("ECT+") and no cleaved substrate ("ECT-") (Fig 1B). Differential expression analysis following RNA-Seq of both populations revealed a small fraction (~0.1%) of genes that were statistically significantly enriched or diminished (S1 Table). A gene that was highly enriched in the most efficient ECT+ population was *AQP3* (false discovery rate adjusted p-value<0.001, Fig 1C).

In parallel, we developed a second, independent screen using ECT reporter assay in the context of gene expression knockdown (Fig 1D and 1E). In this approach, cells expressing the GAL4-UAS reporter element were incubated in the presence of recombinant GAL4 transactivating domain protein (GAL4-TA). For reporter expression to occur, GAL4-TA has to undergo ECT, translocate to the nucleus, and bind to its cognate upstream activating sequence. Using the GAL4-UAS ECT assay, we performed a siRNA knockdown screen in HEK293 cells to identify candidates involved in ECT. AQP3 was once again identified as a potential positive regulator of ECT. This result was confirmed in BMDCs by retroviral transduction of the GAL4-UAS reporter in WT BMDCs followed by transfection of AQP3 siRNA (Fig 1F). A second hit, the AAA-ATPase copper transporter ATP7a, was also confirmed in BMDCs. Since two independent screens identified AQP3 as a positive regulator of ECT, we investigated its function in greater detail.

### AQP3 localizes to phagosomes and transports hydrogen peroxide

In order to examine the attributes of AQP3 important for ECT, we expressed other aquaporin family members and AQP3 mutants in HEK293 cells and performed the β-lactamase ECT assay. Although WT AQP3 increased ECT compared to empty vector, an AQP3 channel mutant (A213W) [24] did not, demonstrating that channel function was required (Fig 2A). We next examined AQP2, an aquaporin family member that has been proposed to localize to early endosomes [25] but that strictly transports water [26], whereas AQP3 also transports glycerol and hydrogen peroxide ($H_2O_2$) [27, 28]. AQP2 overexpression did not increase ECT (Fig 2A), suggesting that aquaporin-mediated transport of a substrate other than water was important. To further test this hypothesis, we generated an AQP3 mutant that was rendered water-specific by changing the glycine at position 203 to a histidine (AQP3 G203H). Aquaporins that transport only water have a histidine at position 203 while aquaporins that have broader substrate specificities exhibit small, uncharged amino acids at position 203 (glycine or alanine) [29]. Indeed, expression of the water-selective AQP3 G203H mutant failed to increase ECT (Fig 2A).

AQP3 and AQP9 are the only aquaporin family members expressed by BMDCs, with both thought to transport similar substrates (water, glycerol, $H_2O_2$) [27]. However, in the RNA-Seq ECT screen, AQP9 was not differentially expressed (S1 Table) and its expression in HEK293 cells did not increase ECT (Fig 2A). Unlike AQP3, AQP9 is thought to be localized primarily at the plasma membrane. In fact, by fluorescence microscopy using RFP-tagged constructs, the distributions of the two aquaporins were distinct, with AQP9 being found at the plasma

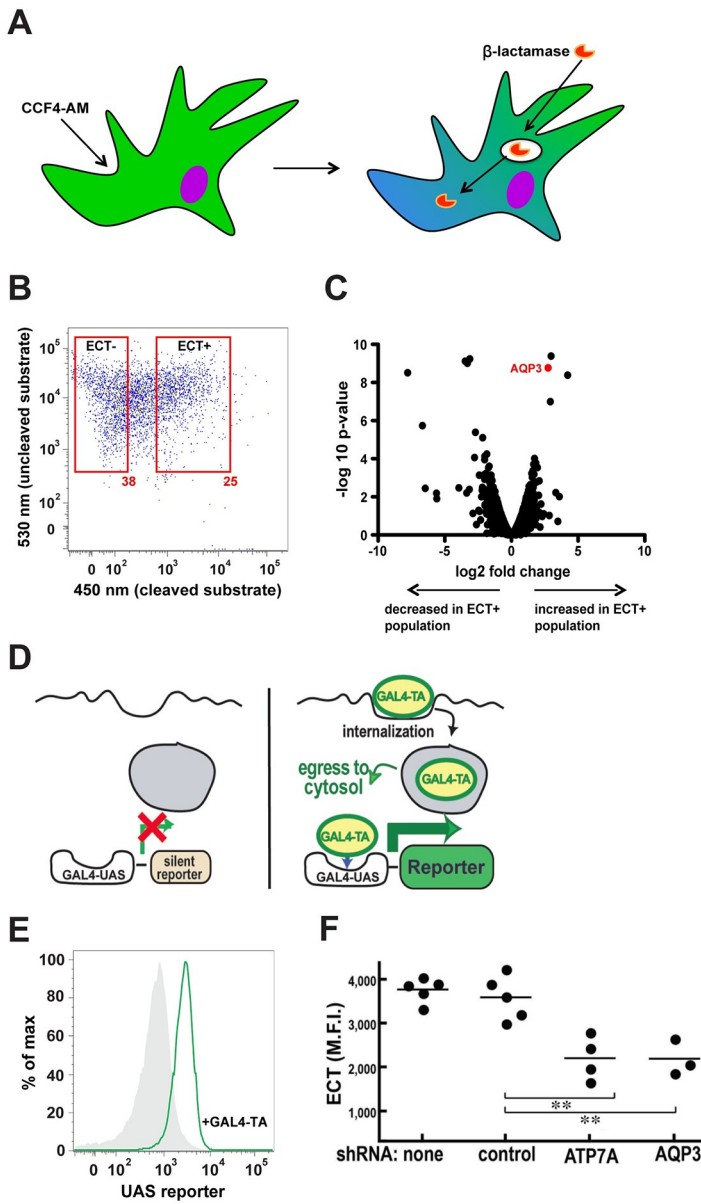

**Fig 1. Two independent functional genomics screens for regulators of ECT uncover *AQP3*.** (**A**) CCF4 ECT assay schematic. (**B**) Representative flow cytometry plot of the sorting of WT BMDCs based on ECT+ or ECT- populations. (**C**) *AQP3* was significantly enriched in the ECT+ population (false discovery rate adjusted p-value<0.001) according to RNA-Seq differential gene expression analysis. (**D**) GAL4-TA ECT assay schematic. (**E**) Representative histograms of the GAL4-UAS reporter cell line demonstrating ECT after incubation with 10 μg/ml GAL4-TA overnight. (**F**) BMDCs expressing the GAL4-UAS reporter cell line were treated with various shRNAs followed by incubation with GAL4-TA overnight. Mean fluorescence intensity is shown. 2 independent experiments were performed. **P<0.01, two-tailed t-test.

membrane with only a small amount of intracellular staining in the perinuclear region. By comparison, AQP3 was localized almost entirely to intracellular compartments (that were not AQP9-positive) (Fig 2B) and could also be readily detected by immuno-electron microscopy on the phagosomal membrane in HEK293 cell transfectants allowed to internalized latex beads (Fig 2C). Immunofluorescence staining of endogenous AQP3 in BMDCs corroborates the

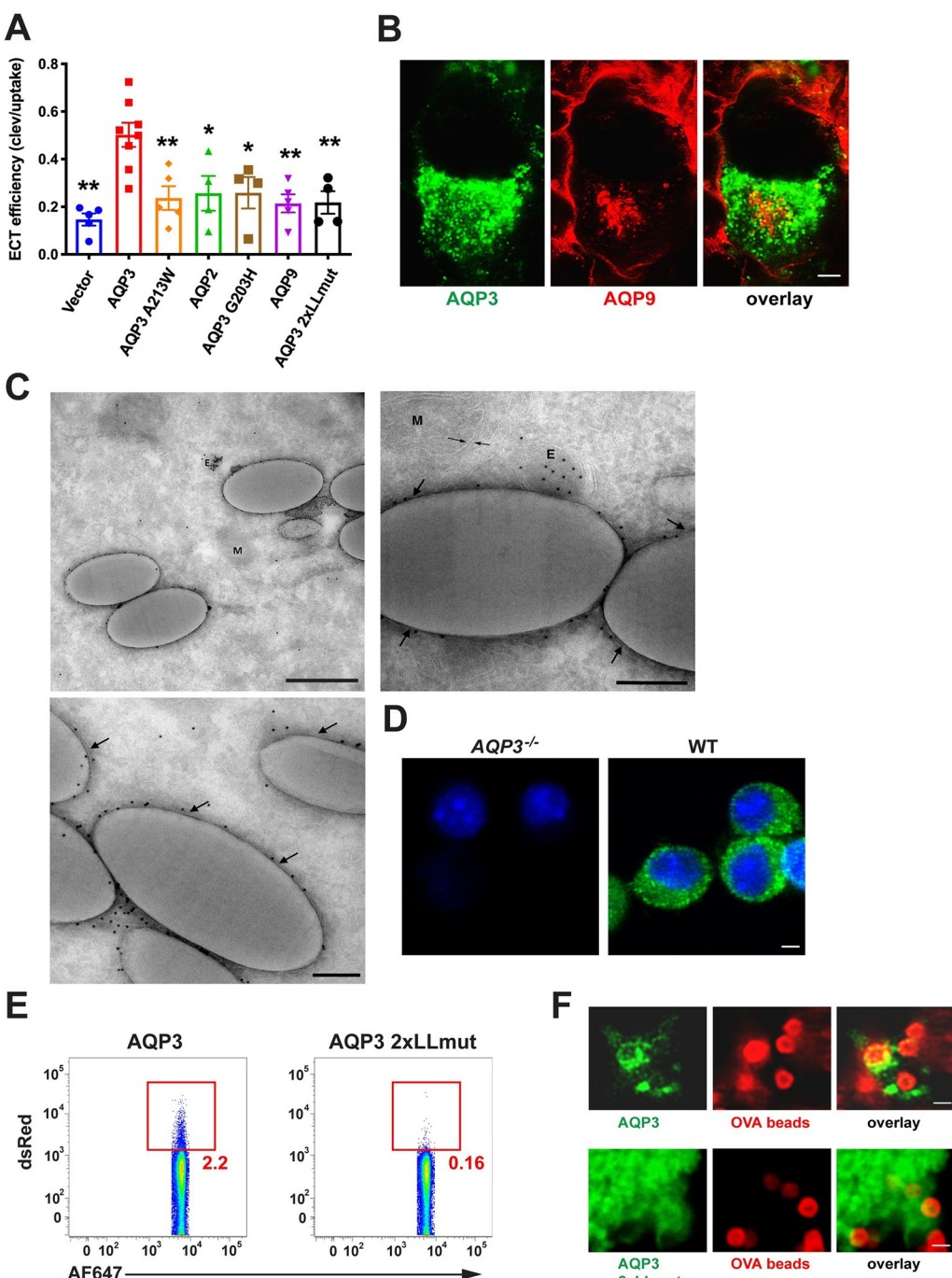

**Fig 2. AQP3 is uniquely positioned among aquaporin family members to regulate ECT.** (**A**) HEK293 cells were transfected with various aquaporin overexpression constructs, flow cytometry-sorted for equivalent expression, and ECT efficiency (described in Methods) was evaluated with 100 μg/ml β-lactamase. Mean ± SEM is shown. (**B**) HEK293 cells were co-transfected with fluorescently-tagged AQP3 and AQP9 expression constructs and 3 days later, fixed and imaged by confocal microscopy. Scale bar: 2 μm. (**C**) HEK293 cells stably expressing fluorescently-tagged AQP3 were incubated with 1 μm latex beads for 2 hours prior to fixation and analysis by immuno-electron microscopy. 3 representative images are shown. Arrows point to phagosomal membrane. E: Endosome. P: Phagosome/latex bead. M: Mitochondria. Scale bar for top left image: 500 nm. Scale bar for top right and bottom left images: 200 nm. (**D**) On day 6 of culture, WT or *AQP3*[-/-] BMDCS were fixed in 4% PFA and permeabilized with 0.05% saponin, followed by detection with anti-AQP3. AQP3 can be seen on or near the plasma membrane and dotting the interior of the cell. Scale bar: 2 μm. (**E**) HEK293 cells were transfected with fluorescently-tagged AQP3 or AQP3 2xLLmut. 3 days later, cells were incubated with fluorescently-

labeled OVA-coated 1 μm beads for 1 hour prior to cell homogenization and analysis by flow cytometry. **(F)** AQP3 2xLLmut does not localize to phagosomes. Stably-transfected HEK293s were incubated with 100:1 (beads:cells) ratio of fluorescently-labeled OVA-coated 1 um tosylactivated Dynabeads and live imaging was initiated immediately. Scale bar: 1 μm. 2–4 independent experiments were performed.

pattern observed in transfected HEK293s, with AQP3 localized on or near the plasma membrane and dotting the interior of the cell (Fig 2D). Since these observations suggested that endolysosomal localization was an important factor in AQP3's ability to facilitate ECT, we generated a "mislocalization mutant" by mutating the two cytoplasmic domain dileucine motifs (AQP3 2xLLmut) previously shown to be important for endolysosomal targeting [16, 30]. By both cell fractionation (Fig 2E) and fluorescence microscopy (Fig 2F) in transfected HEK293s expressing fluorescently-labeled AQP3 constructs and fed latex beads, the AQP3 2xLLmut was found not to reach phagosomes and also failed to increase ECT relative to WT AQP3 (Fig 2A). In summary, the phagolysosomal localization and broader substrate specificity for $H_2O_2$ and glycerol of AQP3 are key to its ability to promote ECT.

Reactive oxygen species (ROS) have been shown to play a role in cross presentation as a regulator of intraphagosomal pH or cysteine protease activity [31, 32]. We next asked if $H_2O_2$ was also important for ECT (images describing the measurement of ECT in BMDCs can be found in S1 Fig). WT BMDCs were incubated in the presence of exogenously applied $H_2O_2$ and found to significantly increase β-lactamase release into the cytosol (Fig 3A). To determine if endogenous ROS/$H_2O_2$ could similarly increase ECT, we loaded BMDCs with the ROS sensor CM-$H_2$DCFDA and treated with viral or bacterial components, using dextran as a negative control. Interestingly, β-lactamase, which is bacterial in origin and a rich source of endotoxin, proved to be a potent stimulator of ROS production, superior to peptidoglycans (PGN) or lipopolysaccharide (LPS) (Fig 3B). Based on staining with the reporter MitoSOX red, this increase appeared largely due to enhanced mitochondrial ROS (mROS) (Fig 3C). In addition, the β-lactamase effect on cellular ROS was significantly diminished in the presence of the mitochondrial electron transport chain inhibitor 2-thenoyltrifluoroacetone (TTFA) (Fig 3B). TTFA also reduced the egress of β-lactamase to the cytosol to a similar extent (Fig 3D), consistent with the possibility that mROS was associated with ECT. In contrast, the phagosome-associated NOX2 system seemed to have little if any role in modulating ECT. BMDCs from $Cybb^{-/-}$ (NOX2-deficient) mice showed no significant decrease in β-lactamase-induced ROS or in its release to the cytosol by ECT (S2 Fig).

These results implicated viral or bacterial products as potent stimuli of mROS in DCs, perhaps not surprising given a recent study demonstrating that pathogen sensing of internalized material was important for mROS production in macrophages [33]. Interestingly, induction of mROS in macrophages was associated with the recruitment of mitochondria to phagosomes and a TRAF6-dependent assembly of the electron transport chain [33]. As TRIF is upstream of TRAF6 in a pathogen-sensing signaling cascade, we investigated whether TRIF deficiency would alter ROS production and ECT. Indeed, BMDCs from $TRIF^{-/-}$ mice were deficient at ECT (Fig 3E), consistent with a previous report [15], and produced less ROS in response to β-lactamase (Fig 3B). Also consistent with the macrophage data, by electron microscopy we often found mitochondria in proximity to latex bead phagosomes (Fig 2C), although this occurred regardless of whether TLR agonists were present.

Since β-lactamase and other microbial agonists appear capable of generating mROS in DCs, we next asked if AQP3 might have a role in the transport of $H_2O_2$ into the lumen of endocytic organelles. Although $H_2O_2$ is often thought of as being membrane-permeable, passive diffusion across membranes is inefficient and is greatly facilitated by the presence of an

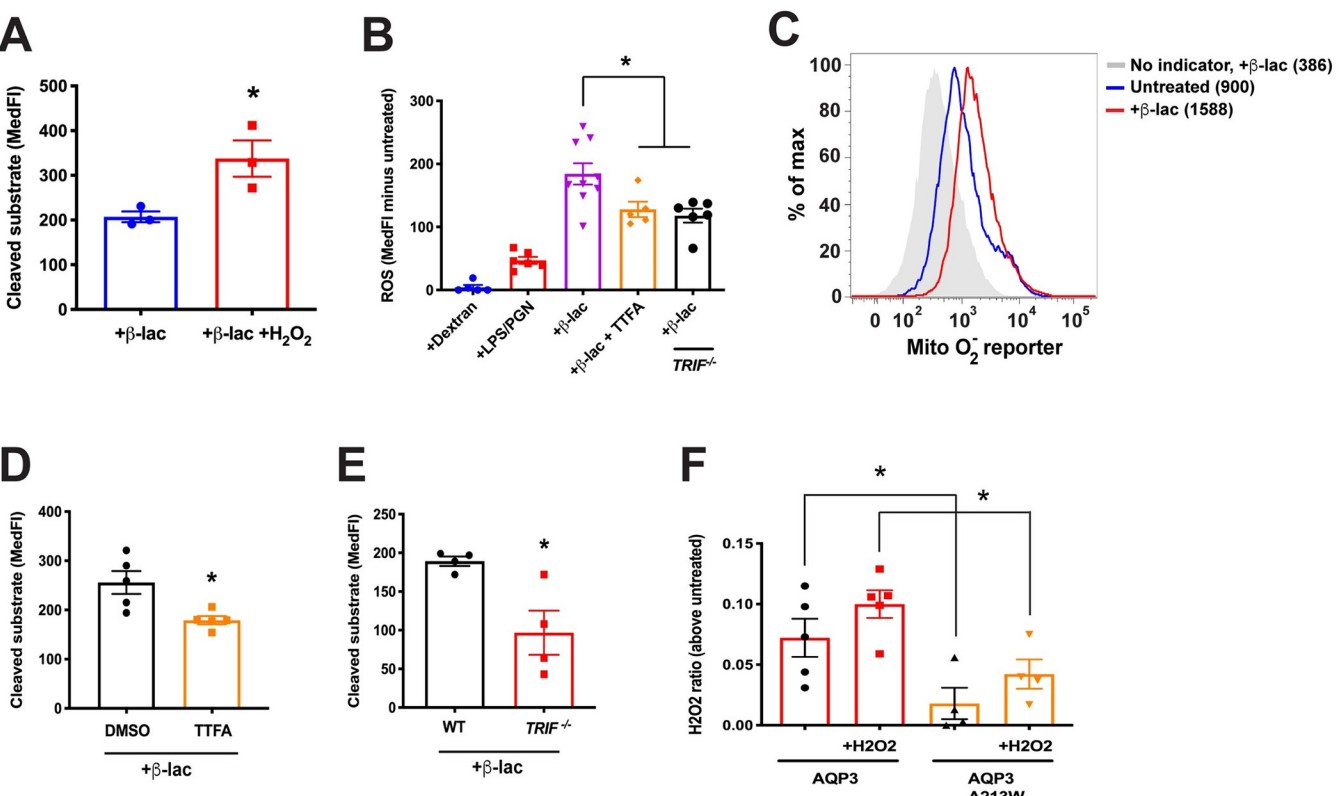

**Fig 3. Cytosolic ROS is important for ECT.** (**A**) ECT was evaluated in WT BMDCs with 100 μg/ml β-lactamase in the presence or absence of 100 μM $H_2O_2$. (**B**) $TRIF^{-/-}$ or WT BMDCs were loaded with the ROS sensor CM-$H_2$DCFDA followed by incubation with 250 μg/ml dextran, 100 μg/ml each PGN and LPS, or 250 μg/ml β-lactamase, in the presence or absence of 1 μM TTFA. (**C**) MitoSOX Red-loaded BMDCs were incubated with 250 μg/ml β-lactamase. MedFI is displayed. (**D**) ECT was evaluated in WT BMDCs with 250 μg/ml β-lactamase in the presence of 1 μM TTFA or equivalent volume DMSO. (**E**) ECT was assessed in $TRIF^{-/-}$ or WT BMDCs with 100 μg/ml β-lactamase. (**F**) HEK293 cells were transfected with the indicated constructs and phagosomal $H_2O_2$ was evaluated with rHyPer in the presence or absence of 100 μM $H_2O_2$. Mean ± SEM is shown. *P<0.05, two-tailed t-test. 2–4 independent experiments were performed.

appropriate aquaporin [34]. We tested this possibility using recombinant HyPer (rHyPer) protein, a highly sensitive and specific $H_2O_2$ sensor [35]. rHyPer was added to the media of HEK293 cells transfected with WT AQP3 or the channel mutant AQP3 A213W. When $H_2O_2$ was added to the media, AQP3-expressing cells had elevated phagosomal $H_2O_2$ as compared to AQP3 channel mutants (Fig 3F). rHyPer can be sensitive to pH changes in addition to $H_2O_2$, therefore, we tested the possibility of a difference in endo-lysosomal pH between HEK293 cells that expressed wild-type AQP3 or AQP3 channel mutants. We observed no difference in endo-lysosomal pH between cells expressing either construct (S3A Fig). Interestingly, although the addition of exogenous $H_2O_2$ increased phagosomal $H_2O_2$ in both conditions, AQP3-expressing cells consistently maintained an elevated level of $H_2O_2$ in the phagosome compared to AQP3 channel mutants. This suggests that AQP3 affects endosomal $H_2O_2$ whether the $H_2O_2$ was generated from endogenous sources or supplied extracellularly.

## AQP3 regulates ECT via endosomal lipid peroxidation

One possible mechanism whereby increased phagosomal $H_2O_2$ might increase ECT is its well-known ability to cause lipid peroxidation and membrane damage [36, 37]. To assess directly whether AQP3 enhanced the extent of lipid peroxidation of phagosomal membranes, we

exposed BMDCs to OVA-coated beads and a fluorescence-based lipid peroxidation indicator C11-bodipy. Cells were then homogenized and the beads were analyzed by flow cytometry. BMDCs from $AQP3^{-/-}$ mice had decreased phagosomal lipid peroxidation, but only when lipid peroxidation was induced with a β-lactamase bacterial stimulus during the bead/C11-bodipy incubation period (Fig 4A).

We next performed a more detailed analysis of the RNA-Seq dataset from the BMDC ECT screen (Fig 1A–1C) to determine if there were other transcriptional alterations that enhance ECT and were consistent with the possibility that ECT and cross presentation were enhanced by $H_2O_2$-induced lipid peroxidation and endosomal/phagosomal lysis. Two features were of particular note: first, a decreased expression of lysosomal proteases, which would slow the deg-radation of internalized antigens in endocytic compartments, favoring antigen presentation [8, 11], and second, a signature of oxidative stress that is associated with enhanced ROS (Fig 4B). Interestingly, the expression profile also exhibited the diminished expression of several iron-sequestering enzymes, which might favor iron-based Fenton reactions that are needed to produce membrane-damaging hydroxyl radicals from $H_2O_2$ [38]. The increased expression of the copper transporter ATP7A (Fig 1F) is also intriguing, given the well-known role of copper (Cu-II) in catalyzing lipid peroxidation [39, 40]. These considerations are consistent with a model where antigen is released from phagosomes via compromised membrane integrity from lipid peroxidation damage.

We therefore asked if another approach to compromising the integrity of endocytic compart-ments might similarly yield an increase in ECT or cross presentation. ISCOMATRIX™ adjuvant is a formulation of saponin, phospholipids, and cholesterol that has demonstrated immune-stim-ulating properties, including the ability to increase CD8+ T cell priming in vitro and in vivo in a MyD88-dependent manner [41]. Although the precise mechanism of action of ISCOMATRIX™ adjuvant is unknown, saponin is an active component suggesting a role for membrane permeabi-lization. We found that ISCOMATRIX™ adjuvant increased ECT (Fig 4C) and cross presentation (Fig 4D) in WT BMDCs. Importantly, the effect on cross presentation was sensitive to the pro-teasome inhibitor epoxomicin, indicating that the ISCOMATRIX™ adjuvant-induced pathway utilized the canonical pathway of antigen cross presentation (Fig 4D). Thus, direct membrane disruption by ISCOMATRIX™ adjuvant enhanced both ECT and antigen presentation, consis-tent with the possibility that ROS-mediated phagosome disruption would perform similarly.

Previous models of cross presentation have proposed that a translocon-like membrane channel such as Sec61 would serve as the route for antigen release and ECT [14, 15]. Although the identity of this putative channel has remained elusive, we devised an experiment to test this mechanism. Since translocation through an ERAD channel absolutely requires protein unfolding [42], we generated a fusion protein of β-lactamase and mammalian dihydrofolate reductase (DHFR). DHFR fusion proteins have been used to evaluate the role of protein unfolding in various settings because DHFR forms a nearly irreversible high affinity complex with methotrexate (MTX) that stabilizes a folded conformation [43]. Even though the β-lacta-mase-DHFR fusion protein was sensitive to MTX inhibition indicating that at least the DHFR moiety was in a folded state (Fig 4E), the β-lactamase fusion protein underwent ECT at the same efficiency as PBS-treated protein (Fig 4F), suggesting that its escape from endocytic com-partments in DCs reflected a more non-specific process, such as coordinated leak, secondary to a loss of membrane integrity.

## AQP3 facilitates antigen cross presentation *in vitro* and *in vivo*

We next evaluated the functional effects of AQP3 modulation in dendritic cells. Of the resident CD11c+ DC populations in mouse, the CD8+ and XCR1+ subsets are considered specialized

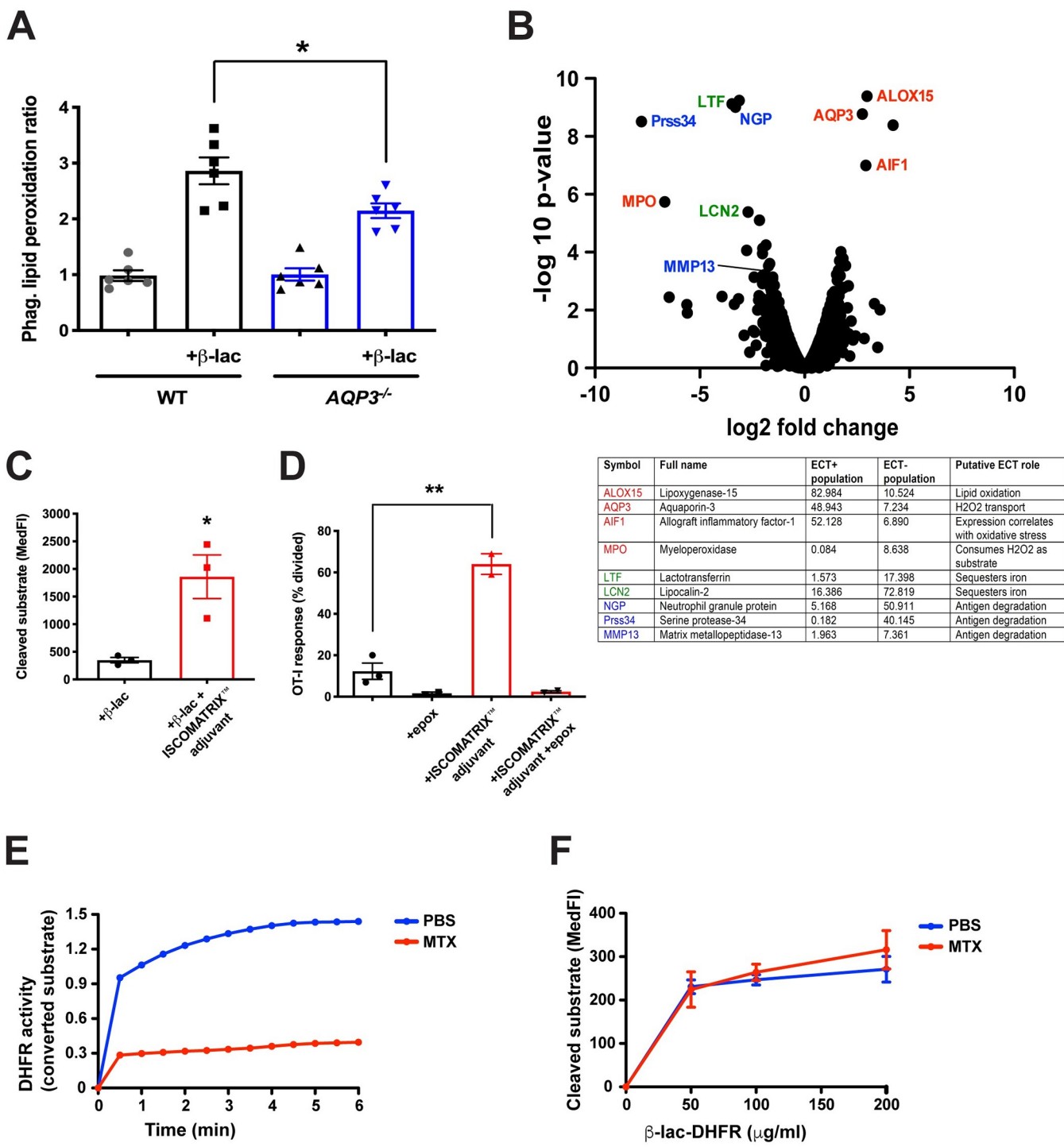

**Fig 4. AQP3 regulates phagosomal lipid peroxidation.** (**A**) *AQP3⁻/⁻* or WT BMDCs were co-incubated with fluorescently-labeled OVA-coated 1 μm beads and a C11-bodipy lipid peroxidation indicator for 1.5 hours, homogenized, and phagosomes were analyzed by flow cytometry. (**B**) Annotated volcano plot of ECT functional genomics screen highlights the connection between ECT and ROS generation and transport, which could potentiate endosomal lipid peroxidation and antigen release. (**C**) ECT was evaluated in WT BMDCs with 0.5 mg/ml β-lactamase in media that did or did not contain ISCOMATRIX™ adjuvant. (**D**) WT BMDCs were incubated with 100 mg/ml OVA in media that did or did not contain ISCOMATRIX™ adjuvant, in the presence or absence of 1 μM epoxomicin, followed by co-incubation with CFSE-labeled OT-I CD8+ T cells for 64 hours. The number of CD8+ T cells that had undergone division as a measure of cross presentation was evaluated by flow cytometry. (**E**) DHFR activity of recombinant β-lactamase-DHFR fusion protein was measured after incubation with 500 nM MTX or equivalent volume PBS. (**F**) The ability of increasing amounts of recombinant β-lactamase-DHFR fusion protein to undergo ECT was evaluated in WT BMDCs following treatment with 500 nM MTX or equivalent volume PBS. *P<0.05, *P<0.01, two-tailed t-test. 2–3 independent experiments were performed.

for cross presentation [44], with recent data implicating XCR1+ DCs as the most efficient cross presenting cells *in vivo* [45–47]. Previous studies have suggested that a significant portion of cross presentation efficiency in certain DC subsets results from limited degradation and increased ECT of internalized antigen [10, 48]. Therefore, we hypothesized that the more efficient the cross presenting cell, the more sensitive it would be to ECT perturbations induced by deleting AQP3. As shown in Fig 5A–5C, *AQP3*[-/-] BMDCs, as well as both CD8+ and XCR1 + splenic DCs, exhibited reduced ECT compared to WT controls. However, the magnitude of the difference between WT and *AQP3*[-/-] was greatest in XCR1+ DCs (Fig 5C). The decrease was not explained by reduced β-lactamase uptake by the mutant DCs (S3B and S3C Fig). These results demonstrate a reliance on AQP3 for efficient ECT in all DCs tested, with the largest difference observed in the XCR1+ subset that is specialized for cross presentation.

As AQP3 may play a partial role in the controlled disruption of endosomal compartments and thus cytosolic release of internalized antigen, we next asked if it was involved in antigen cross presentation in functionally relevant settings. We first overexpressed AQP3 in BMDCs by viral transduction and performed cross presentation assays. AQP3 overexpression in BMDCs increased cross presentation to both soluble and antibody-conjugated antigen (Fig 5D and 5E). Importantly, presentation of pre-processed peptide was not affected in AQP3-overexpressing cells (Fig 5D); antigen uptake (S4A and S4B Fig) and MHC class II antigen presentation were also unchanged (Fig 5F). In contrast, AQP3 shRNA knockdown in WT BMDCs decreased ECT (Fig 5G) and reduced cross presentation (Fig 5H), with no measurable difference in antigen uptake (S4C and S4D Fig).

To explore the role of AQP3 modulation *in vivo*, we evaluated the ability of *AQP3*[-/-] mice to control an infection with lymphocytic choriomeningitis virus (LCMV), a process that relies on efficient cross presentation of viral antigens [49–51]. *AQP3*[-/-] mice were more susceptible to LCMV challenge as indicated by higher viral titer in the kidney 8 days post infection (Fig 5I). In addition, *AQP3*[-/-] mice displayed a partial but significant reduction in the generation of LCMV antigen-specific IFNγ+ CD8+ T cells (Fig 5J), again consistent with impaired cross presentation.

In order to follow up on the viral challenge results, we next asked whether AQP3 deletion altered the ability to prime naïve CD8+ T cells in response to exogenous antigen captured and cross presented by APCs. In these studies, WT or *AQP3*[-/-] mice were immunized intraperitoneally with recombinant HER-2/neu antigen and subsequent analysis of splenocytes 1 week later revealed a significant reduction in newly generated HER-2/neu-specific tetramer-positive CD8+ T cells in *AQP3*[-/-] mice (Fig 5K). In contrast, the CD4+ T cell-dependent antigen-specific antibody response was similar in *AQP3*[-/-] mice compared to controls (S4E Fig). Taken together, these data demonstrate that AQP3 can affect cross presentation *in vivo*, with no observable effect on CD4+ T cell-dependent/MHC class II-based antigen presentation.

## Discussion

This study provides new insight into the long-standing question as to how internalized material escapes from endocytic compartments. Although ECT is not wholly dependent on AQP3, expression of functional AQP3 capable of transporting $H_2O_2$ into endocytic compartments increased the efficiency of ECT, while AQP3 deletion decreased ECT. Our data are inconsistent with a role for specific, unknown protein channels; however they are not inconsistent with the possibility that the ER itself has a role in these events, as has been suggested many times. Indeed, in yeast and other non-immune cells, there is increasing evidence of direct contact between ER and endosomal elements, which are important for endosomal sorting functions [52].

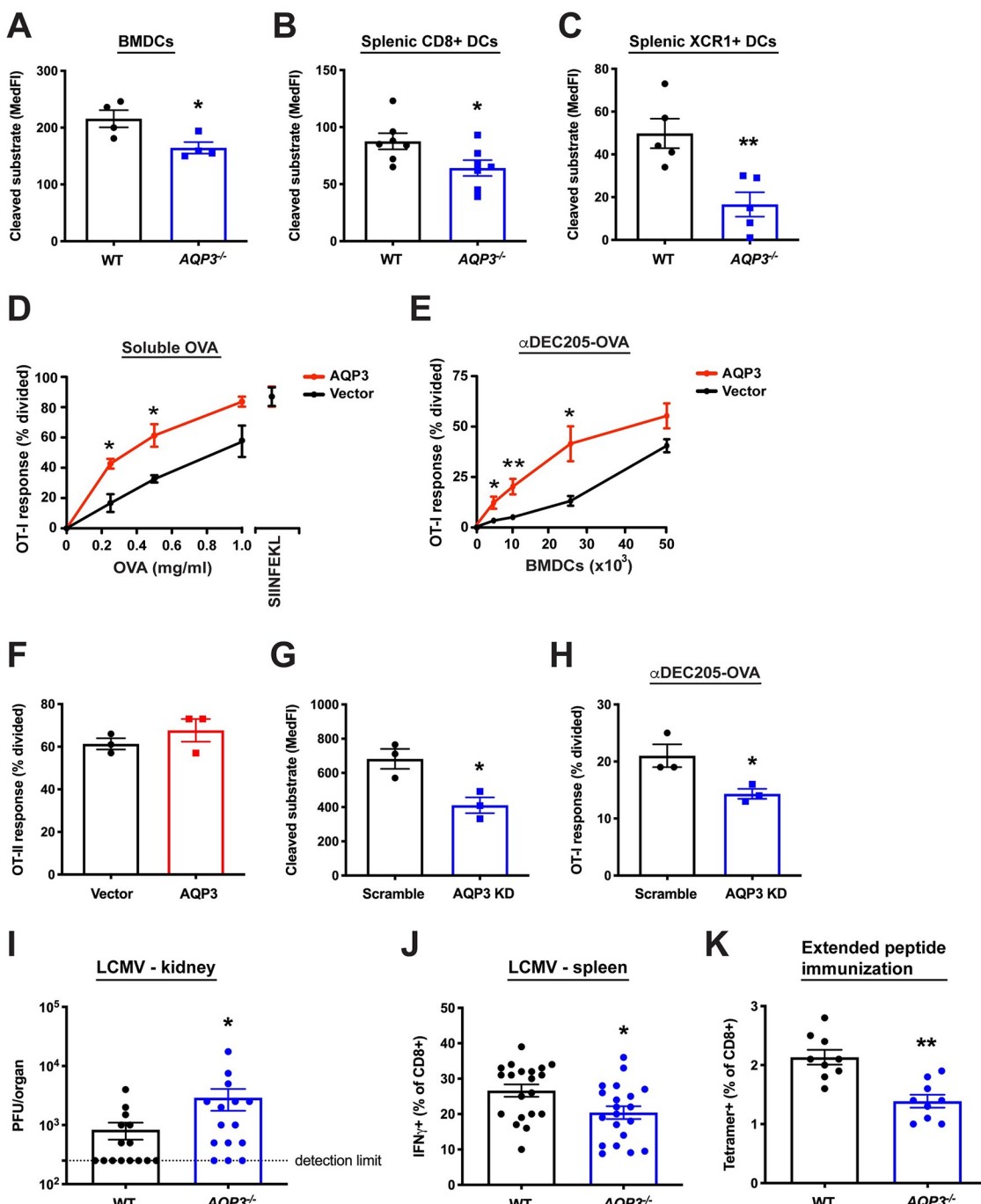

**Fig 5. AQP3 is involved in cross presentation efficiency *in vitro* and *in vivo*.** (**A**) ECT was evaluated in *AQP3*⁻/⁻ or WT BMDCs with 50 μg/ml β-lactamase. (**B-C**) Splenic CD11c+ DCs from *AQP3*⁻/⁻ or WT mice were isolated and ECT was evaluated in the CD11c+CD8+ population (**B**) or the CD11c+XCR1+ population (**C**) with 0.5 mg/ml β-lactamase. (**D**) BMDCs overexpressing AQP3 by viral transduction were incubated with increasing amounts of OVA or SIINFEKL peptide followed by co-incubation with CFSE-labeled OT-I CD8+ T cells. (**E**) BMDCs overexpressing AQP3 by viral transduction were incubated with 1 μg/ml αDEC205-OVA followed by co-incubation with CFSE-labeled OT-I CD8+ T cells. (**F**) BMDCs were incubated with 250 μg/ml OVA followed by co-incubation with CFSE-labeled OT-II CD4+ T cells. (**G**) ECT was evaluated in BMDCs expressing an AQP3 shRNA knockdown construct by viral transduction with 100 μg/ml β-lactamase. AQP3 shRNA knockdown was calculated to be 93% effective as measured by real-time PCR. (**H**) AQP3 KD cells were incubated with 1 μg/ml αDEC205-OVA followed by co-incubation with CFSE-labeled OT-I CD8+ T cells. (**I,J**) *AQP3*⁻/⁻ and WT control mice were infected with LCMV Armstrong. 8 days later, viral titer in kidney was assessed (**I**) and splenocytes were stimulated with LCMV NP peptide to determine the generation of LCMV-specific CD8 + T cell clones (**J**). Each dot represents an individual mouse. (**K**) *AQP3*⁻/⁻ and WT control mice were injected with rat HER-2/neu

extended peptide and adjuvant. 1 week later, splenocytes were harvested and the generation of antigen-specific CD8+ T cell clones was evaluated using a HER-2/neu tetramer. Mean ± SEM is shown. *P<0.05, **P<0.01, two-tailed t-test or Mann-Whitney U test. 2–3 independent experiments were performed.

Additionally, we found that a variety of pathogenic stimuli enhanced production of mROS, thereby providing additional substrate for AQP3. This process was partially dependent on TRIF, which sits upstream of TRAF6, a molecule critically involved in ROS signaling [33, 53]. These results provide further support for a link between pathogen sensing and increased antigen presentation efficiency, and suggest that ECT is another step in the cross presentation pathway that to some degree is modulated by pathogenic stimuli.

Although we did observe a decreased anti-viral response and reduced ability to cross present extended peptide in the $AQP3^{-/-}$ mice, we no longer saw a consistent cross presentation defect in $AQP3^{-/-}$ once the mice were crossed to the C57BL/6 background and immunized with OVA-expressing necroptotic cells [54]. In addition, given that the results presented here support an ECT model in which cargo egress occurs as a result of compromised membrane integrity from endosomes that have experienced lipid peroxidation and ostensibly is not antigen specific, we also tested if AQP3 deletion had any effect on STING pathway activation in macrophages [55]. In these series of experiments, irradiated cells transfected with STING agonists were incubated with bone marrow-derived macrophages and indicators of intracellular STING pathway activation were measured, with no difference observed between $AQP3^{-/-}$ and WT control cells. The reason behind the lack of an effect of AQP3 deletion in these experiments is unclear, although the contribution of AQP3 to ECT may be most pronounced when assessed in the context of soluble protein antigen, as both functional genomics screens that identified AQP3 relied on this antigen delivery method.

Finally, while AQP3 is important in regulating ECT efficiency, it should be remembered that it is one of a number of specializations that together confer DCs with an enhanced capacity for cross presentation. Like AQP3, none of the specializations such as reduced levels of lysosomal proteases to preserve antigen, activation of mROS in response to TLR agonists, or expression of IL-12 to prime CD8+ T cells are DC specific, suggesting that ECT is a process that can occur in most cell types, albeit with reduced efficiency. Indeed, a variety of internalized substances gain access to cytosolic compartments to facilitate surveillance by the innate immune system [16, 56]. Viewed in this light, it is possible ECT is an evolutionarily conserved mechanism to enable all cells to sample pathogenic endosomal material and engage cytosolic sensors while maintaining a relatively safe topological barrier between the pathogen and the interior of the cell. Perhaps DCs and other antigen presenting cells have co-opted this process for cross presentation by rendering it more efficient and subject to some level of regulation by linking it to mROS generation.

## Supporting information

**S1 Fig. Representative images demonstrating the measurement of ECT in BMDCs.** Cleaved substrate is the median fluorescence intensity of the 450 nM channel.
(PDF)

**S2 Fig. $Cybb^{-/-}$ BMDCs are not defective at cellular ROS production or ECT.** (**A**) $Cybb^{-/-}$ or WT BMDCs were loaded with the ROS sensor CM-H$_2$DCFDA followed by incubation with 100 µg/ml β-lactamase. (**B**) ECT was evaluated with 100 µg/ml β-lactamase.
(PDF)

**S3 Fig. AQP3 overexpression does not affect endosomal pH and β-lactamase phagocytosis is not different between *AQP3*$^{-/-}$ and WT DCs.** (**A**) Transfected and sorted HEK293s were incubated with LysoSensor Green DND-189 for 30 minutes followed by analysis by flow cytometry. The greater the LysoSensor fluorescence intensity, the lower the pH. MedFI is displayed. (**B**) Phagocytosis of 50 μg/ml fluorescently-labeled β-lactamase in BMDCs from *AQP3*$^{-/-}$ or WT control mice. (**C**) Phagocytosis of 0.5 mg/ml fluorescently-labeled β-lactamase in splenic CD11c+CD8+ DCs isolated from *AQP3*$^{-/-}$ or WT control mice. (**D**) Phagocytosis of 0.5 mg/ml fluorescently-labeled β-lactamase in splenic CD11c+XCR1+ DCs isolated from *AQP3*$^{-/-}$ or WT control mice. MedFI is displayed.
(PDF)

**S4 Fig. AQP3 overexpression or knockdown in transduced BMDCs does not alter uptake and *AQP3*$^{-/-}$ mice are not defective in the production of anti-OVA Igs.** (**A**) Phagocytosis of 50 μg/ml fluorescently-labeled OVA. (**B**) Phagocytosis of 1 μg/ml fluorescently-labeled αDEC205 (protocol described in Methods). (**C**) Phagocytosis of 100 μg/ml fluorescently-labeled β-lactamase. (**D**) Phagocytosis of 1 μg/ml fluorescently-labeled αDEC205. MedFI is displayed. (**E**) *AQP3*$^{-/-}$ or WT control mice were injected i.p. with OVA in CFA. 10 days later, serum was collected and analyzed for anti-OVA Igs.
(PDF)

**S1 Table. Full dataset for ECT functional genomics screen in WT BMDCs.**
(XLS)

**S1 File. The ARRIVE essential 10: Author checklist.**
(PDF)

# Acknowledgments

We thank Suzanne van Dijk for excellent technical EM support and Mohammad Samie for critical reading of the manuscript.

# Author Contributions

**Conceptualization:** Sam C. Nalle, Ira Mellman.

**Data curation:** Sam C. Nalle, E. Sergio Trombetta.

**Formal analysis:** Sam C. Nalle, E. Sergio Trombetta, Ira Mellman.

**Investigation:** Sam C. Nalle, Rosa Barreira da Silva, Hua Zhang, Markus Decker, Cecile Chalouni, George Posthuma, Ann de Mazière, Sebastian J. Fleire.

**Resources:** Adriana Baz Morelli, Alan S. Verkman, E. Sergio Trombetta, Matthew L. Albert, Ira Mellman.

**Supervision:** Min Xu, Judith Klumperman, E. Sergio Trombetta, Matthew L. Albert, Ira Mellman.

**Writing – original draft:** Sam C. Nalle.

**Writing – review & editing:** Rosa Barreira da Silva, E. Sergio Trombetta, Matthew L. Albert, Ira Mellman.

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
