## [Decision Letter · Decision Letter 0]

7 Sep 2020

PONE-D-20-25085

Aquaporin-3 regulates endosome-to-cytosol transfer via lipid peroxidation for cross presentation

PLOS ONE

Dear Dr. Nalle,

Thank you for submitting your manuscript to PLOS ONE. As you will see, both reviewers have requested a few minor revisions that you will need to operate before a final decision can be taken on your submitted manuscript.

We look forward to receiving your revised manuscript.

Kind regards,

Ludger Johannes

Academic Editor

PLOS ONE

Journal Requirements:

2. As part of your revision, please complete and submit a copy of the ARRIVE Guidelines checklist, a document that aims to improve experimental reporting and reproducibility of animal studies for purposes of post-publication data analysis and reproducibility: https://www.nc3rs.org.uk/arrive-guidelines. Please include your completed checklist as a Supporting Information file. Note that if your paper is accepted for publication, this checklist will be published as part of your article.

3. To comply with PLOS ONE submissions requirements, in your Methods section, please provide additional information on the animal research and ensure you have included details on (1) methods of sacrifice, (2) methods of anesthesia and/or analgesia, and (3) efforts to alleviate suffering.

4.Thank you for stating the following in the Competing Interests section:

[The authors have declared that no competing interests exist.].   

We note that one or more of the authors are employed by a commercial company: CSL Ltd., Genentech, and Boehringer Ingelheim

Reviewers' comments:

Reviewer's Responses to Questions

**Comments to the Author**

1. Is the manuscript technically sound, and do the data support the conclusions?

Reviewer #1: Yes

Reviewer #2: Yes

2. Has the statistical analysis been performed appropriately and rigorously? 

Reviewer #1: I Don't Know

Reviewer #2: Yes

3. Have the authors made all data underlying the findings in their manuscript fully available?

Reviewer #1: Yes

Reviewer #2: Yes

4. Is the manuscript presented in an intelligible fashion and written in standard English?

Reviewer #1: Yes

Reviewer #2: Yes

5. Review Comments to the Author

Reviewer #1: In this work, the authors address the mechanisms of endosome-to-cytosol transfer of antigen, and identify Aquaporin-3 as a molecule involved in this process.

While the study is well conducted and the conclusions well supported by the data shown, there are a few points to improve before the manuscript is suitable for publication.

1. The number of independent experiments performed must be indicated in all figures, and the authors need to indicate what is represented (mean ? median ?) and what errors bars are (SEM ? SD ?).

2. In figure 1C, it would help to add a legend on the volcano plot (what is represented on the left hand side ? right hand side ?).

3. In figure 1F, it is not clear how ECT is measured. This needs to be better explained in the text.

4. It is very difficult to understand what is shown in figure 2E. The authors need to better explain this experiment in the text. The legend for the y axis of the plots is also missing.

5. In figure 3A, the authors should show a representative example of FACS data. It would also help understand how the “cleaved substrate” metrics is measured.

6. In figure 3F, it is not clear how the authors can conclude on the levels of H2O2 in the phagosomes. Do they assume that the sensor which is added in the medium only reaches the phagosomes of the cells ?

7. In figure 4A, it is difficult to understand what is represented. The authors should show a representative example of FACS data.

8. In figure 4D, the authors should indicate in the legend at what time point the cell proliferation was analyzed.

9. In figure 5, it is not clear to me why the authors analyze CD8+ DC and XCR1+ DC separately. These cells both represent cDC1. How did they separate them ? The gating strategy should be shown.

10. Is AQP3 differentially expressed between mouse cDC1 and cDC2 ? This information can probably be found in ImmGen database and could add relevant information regarding the specialisation of cDC1 for cross-presentation.

Reviewer #2: This is an elegant study on the highly important topic of antigen cross-presentation by dendritic cells of the immune system. In this study, two independent genetic screens were performed for identifying factors mediating ECT, a process well-understood to be important for antigen cross-presentation. Both screens identified the aquaporin AQP3 as a factor in ECT. A subsequent set of experiments with knockdown/rescue of AQP3, exogenously addition of H2O2, phagosomal H2O2 sensors, and mitochondrial ROS measurements then led to conclusion that mitochondria (which are positioned close to the phagosomal membrane) produce H2O2 which can diffuse through AQP3 into the lumen of mitochondria. Experiments with a lipid peroxidation probe showed that this correlates with the oxidation of phagosomal membranes. As lipid oxidation is well-known to cause membrane disruption, the authors propose a highly novel model where AQP3 translocates mitochondrial ROS into antigen-containing phagosomes which disrupts the phagosomal membrane by oxidation and thereby leads to ECT. The involvement of AQP3 in antigen cross-priming is confirmed in 2 in vivo using mouse models. The authors also show that the mitochondrial ROS production is enhanced following pathogen recognition via the adapter protein TRIF.

Altogether, the data is very convincing, and the study well designed and presented. The model proposed by the authors is radically new, and provides an explanation to the observation that large, folded proteins and other macromolecules can translocate from the lumen of endo/phagosomes into the cytosol. Of course, as with any new finding, the model also introduces several new questions that need to be addressed in future investigations, particularly why ROS have to translocate to the inside of phagosomes and cannot oxidize the phagosomal membrane from the cytosolic side. Thus, I consider this an important contribution to the antigen cross-presentation field and I expect that this study will attract a lot of attention. I recommend publication and only have some minor comments/suggestions that the authors might want to consider:

1-The actual data from the siRNA screen where AQP3 was found to mediate ECT should be shown somewhere, perhaps as a supplementary figure.

2-In figure 2A, it is shown that AQP2 cannot increase ECT. However, does AQP2 locate to the phagosomal membrane like AQP3? Note that while it would be good to have this information, as it shows the validity of AQP2 as a control, I consider it not strictly essential as the authors provide additional controls with the water-specific G203H mutant of AQP3 and a mutant with a disrupted dileucine motif.

3-In figure 2E (flow cytometry) the label for the y-axis is missing.

4-Maybe this is too speculative, but in plants aquaporins are well known to directly coordinate membrane contact sites between organelles (e.g. https://doi.org/10.1111/nph.16743). Maybe AQP3 itself could coordinate the contact between mitochondria and phagosomes?

5-In fig S1, the authors show that the NADPH oxidase NOX2 is not involved in ECT. This contradicts previous findings from several groups. Can the authors speculate on the reason for this discrepancy?

6. PLOS authors have the option to publish the peer review history of their article (what does this mean?). If published, this will include your full peer review and any attached files.

Reviewer #1: No

Reviewer #2: **Yes: **Geert van den Bogaart

---

## [Author Response · Author response to Decision Letter 0]

28 Oct 2020

Please refer to the Response to Reviewers document.

---

## [Decision Letter · Decision Letter 1]

4 Nov 2020

Aquaporin-3 regulates endosome-to-cytosol transfer via lipid peroxidation for cross presentation

PONE-D-20-25085R1

Dear Dr. Nalle,

We’re pleased to inform you that your manuscript has been judged scientifically suitable for publication and will be formally accepted for publication once it meets all outstanding technical requirements.

Kind regards,

Ludger Johannes

Academic Editor

PLOS ONE

Additional Editor Comments (optional):

Reviewers' comments:

Reviewer's Responses to Questions

**Comments to the Author**

1. If the authors have adequately addressed your comments raised in a previous round of review and you feel that this manuscript is now acceptable for publication, you may indicate that here to bypass the “Comments to the Author” section, enter your conflict of interest statement in the “Confidential to Editor” section, and submit your "Accept" recommendation.

Reviewer #1: All comments have been addressed

Reviewer #2: All comments have been addressed

2. Is the manuscript technically sound, and do the data support the conclusions?

Reviewer #1: (No Response)

Reviewer #2: Yes

3. Has the statistical analysis been performed appropriately and rigorously? 

Reviewer #1: (No Response)

Reviewer #2: Yes

4. Have the authors made all data underlying the findings in their manuscript fully available?

Reviewer #1: (No Response)

Reviewer #2: Yes

5. Is the manuscript presented in an intelligible fashion and written in standard English?

Reviewer #1: (No Response)

Reviewer #2: Yes

6. Review Comments to the Author

Reviewer #1: (No Response)

Reviewer #2: The authors have addressed all my comments. This study demonstrates a highly novel mechanism of antigen cross-presentation by immune cells, and I recommend publication.

7. PLOS authors have the option to publish the peer review history of their article (what does this mean?). If published, this will include your full peer review and any attached files.

Reviewer #1: No

Reviewer #2: **Yes: **Geert van den Bogaart

---

## [Editor Report · Acceptance letter]

6 Nov 2020

PONE-D-20-25085R1 

Aquaporin-3 regulates endosome-to-cytosol transfer via lipid peroxidation for cross presentation 

Dear Dr. Nalle:

I'm pleased to inform you that your manuscript has been deemed suitable for publication in PLOS ONE. Congratulations! Your manuscript is now with our production department. 

Kind regards, 

on behalf of

Dr. Ludger Johannes 

Academic Editor

PLOS ONE